

# 16S rRNA metagenomic analysis of the bacterial community associated with turf grass seeds from low moisture and high moisture climates

Qiang Chen, William A. Meyer, Qiuwei Zhang and James F. White

Department of Plant Biology, Rutgers, The State University of New Jersey, New Brunswick, NJ, USA

## ABSTRACT

Turfgrass investigators have observed that plantings of grass seeds produced in moist climates produce seedling stands that show greater stand evenness with reduced disease compared to those grown from seeds produced in dry climates. Grass seeds carry microbes on their surfaces that become endophytic in seedlings and promote seedling growth. We hypothesize that incomplete development of the microbiome associated with the surface of seeds produced in dry climates reduces the performance of seeds. Little is known about the influence of moisture on the structure of this microbial community. We conducted metagenomic analysis of the bacterial communities associated with seeds of three turf species (*Festuca rubra*, *Lolium arundinacea*, and *Lolium perenne*) from low moisture (LM) and high moisture (HM) climates. The bacterial communities were characterized by Illumina high-throughput sequencing of 16S rRNA V3–V4 regions. We performed seed germination tests and analyzed the correlations between the abundance of different bacterial groups and seed germination at different taxonomy ranks. Climate appeared to structure the bacterial communities associated with seeds. LM seeds vectored mainly Proteobacteria (89%). HM seeds vectored a denser and more diverse bacterial community that included Proteobacteria (50%) and Bacteroides (39%). At the genus level, *Pedobacter* (20%), *Sphingomonas* (13%), *Massilia* (12%), *Pantoea* (12%) and *Pseudomonas* (11%) were the major genera in the bacterial communities regardless of climate conditions. *Massilia*, *Pantoea* and *Pseudomonas* dominated LM seeds, while *Pedobacter* and *Sphingomonas* dominated HM seeds. The species of turf seeds did not appear to influence bacterial community composition. The seeds of the three turf species showed a core microbiome consisting of 27 genera from phyla Actinobacteria, Bacteroidetes, Patescibacteria and Proteobacteria. Differences in seed-vectored microbes, in terms of diversity and density between high and LM climates, may result from effects of moisture level on the colonization of microbes and the development of microbe community on seed surface tissues (adherent paleas and lemmas). The greater diversity and density of seed vectored microbes in HM climates may benefit seedlings by helping them tolerate stress and fight disease organisms, but this dense microbial community may also compete with seedlings for nutrients, slowing or modulating seed germination and seedling growth.

Corresponding author
James F. White,
jwhite3728@gmail.com

# INTRODUCTION

Plants bear numerous microbes that influence their nutrition, development, stress responses and phenotypes (*Hardoim et al., 2015*; *Henning et al., 2016*; *White et al., 2014*, *2018*). Plant-associated microbes generally come from the surrounding environment; however, *Johnston-Monje & Raizada (2011)* showed that corn seeds vectored a diverse array of microbes. Further, *Johnston-Monje et al. (2016)* found that the rhizospheres of corn seedlings were composed of microbes that originated both from seeds and bacteria recruited from soils. These seed-vectored bacteria can influence germination and share a mutualistic association with the host seedlings (*Cruz, Yañez-Ocampo & Wong-Villarreal, 2014*; *Shaik & Thomas, 2019*; *Somova et al., 2001*; *Zhu et al., 2017*). Without seed-vectored bacteria, seedlings may lose gravitropic response of roots, fail to develop root hairs, and are more susceptible to soil-borne pathogens (*Verma et al., 2017*, *2018*).

Some turf grasses possess fungal endophytes of ascomycete genus *Epichloë* that provide resistance to pathogens and insects, and increase abiotic stress tolerance in the host (*Bultman & Bell, 2003*; *Clay, 1990*; *Meyer, Torres & White, 2012*; *White, 1987*). Turf breeders have long been employing these environmentally safe endophytes to enhance turfgrass performance and stress tolerance (*Meyer, Torres & White, 2012*). Another important microbe resource, the seed-transmitted bacterial communities of turf grasses, are yet to be fully explored. Many of these bacteria are vectored on the surfaces of seeds and embedded within dried plant tissues (paleas and lemmas) that adhere tightly to seed surfaces (*White et al., 2019*). During seed germination some of these seed-surface microbes are activated and they externally and internally colonize seedling roots at the root tip meristems, becoming intercellular and intracellular endophytes in the emergent seedling roots (*Verma et al., 2017*, *2018*; *White et al., 2018*). In this study, we employed Illumina HTS and 16S metagenomic analysis to investigate the bacterial community associated with cool-season turfgrass seeds produced in low moisture (LM) and high moisture (HM) climates (*Pace et al., 1986*; *Riesenfeld, Schloss & Handelsman, 2004*). We also evaluated the potential influence of the bacterial community on seed germination rates and seedling growth rates. The results showed that HM seeds vectored a denser and more diverse bacterial community than LM seeds. Also, bacterial groups at different taxonomic ranks correlated with the seed germination rate and time.

# MATERIALS AND METHODS

## Total DNA extraction from seeds of cool-season turfgrasses

Seeds of 27 cool-season turf cultivars were obtained from DLF Pickseed USA (Table S1). All varieties were produced from 2011 to 2015 at either Store Hedinge, Denmark or Les Alleuds, France. Based on the precipitation data collected from The National Oceanic and Atmospheric Administration, the seeds were classified into LM seeds (annual precipitation <750 mm) and HM seeds (annual precipitation >750 mm). With this classification, five samples were classified as LM seeds while 22 samples as HM seeds. A total of 100 mg of seeds of each turf cultivar were weighed out and washed with water for three times, 30 s each time to

remove the dirt. The cleaned seeds were then ground into powder with a sterilized mortar and pestle for total DNA extraction. The DNA extraction was conducted with DNeasy® PowerSoil® Kit (QIAGEN, Hilden, Germany) following the manufacturer's instructions. This PowerSoil® Kit was chosen due to its versatility with diverse sample types. The concentration of extracted DNA was measured with The NanoDrop® ND-1000 Spectrophotometer and normalized to five ng/µl for the library preparation.

## Library preparation and sequencing

The preparation of DNA libraries for each sample followed the Illumina guidelines. By using 12.5 ng of the normalized DNA from turf seeds as the template, V3–V4 hypervariable regions of bacterial 16S rRNA gene were amplified with the primer pair, S-D-Bact-0341-b-S-17 (5′-CCTACGGGNGGCWGCAG-3′) and S-D-Bact-0785-a-A-21 (5′-GACTACHVGGGTATCTAATCC-3′) fuzed with Illumina overhang forward adapters (5′-TCGTCGGCAGCGTCAGATGTGTATAAGAGACAG-3′) and reverse adapter (5′-GTCTCGTGGGCTCGGAGATGTGTATAAGAGACAG-3′), respectively (*Klindworth et al., 2013*). PCR clean-ups were conducted to purify the 16S V3–V4 amplicons away from free primers and primer dimers. Nextera XT index primers were then used for the index PCR and PCR clean-ups were performed again to generate the final library. The generated 16S V3–V4 region library was paired-end sequenced (2 × 300 bp) on an Illumina MiSeq platform in the Genome Cooperative Sequencing Facility, School of Environmental and Biological Sciences at Rutgers.

## Bacterial community structure analysis

The collected sequencing data in FASTQ format was processed and analyzed with the QIIME2 software suite (*Caporaso et al., 2010*). The raw Illumina reads were imported into QIIME2 with "Casava 1.8 paired-end demultiplexed fastq" method, and then denoised and filtered with *dada2* pipeline to remove noisy and chimeric sequences, construct denoised paired-end sequences, and dereplicate them (*Callahan et al., 2016*). De novo clustering was then carried out with *VSEARCH* plugin at 99% identity to generate Operational Taxonomic Units (OTUs) (*Rognes et al., 2016*). The taxonomy assignment of OTUs was performed by using *feature-classifier* against the SILVA 1.28 database (released 29 September 2016). After removing mitochondria and chloroplast sequences, the filtered data were aligned with *mafft* program and *fasttree* method to generate rooted and unrooted phylogenetic trees (*Price, Dehal & Arkin, 2010*). All core metrics used in alpha and beta diversity analysis were computed based on the rooted phylogenetic tree. Alpha diversity (intra group diversity) was calculated with the observed OTUs and Faith's phylogenetic diversity (*Faith, 1992*) at the sample depth of 1,000 reads to normalize the variance and this excluded four samples (three HM samples and one LM sample), leaving four LM samples and 19 HM samples. The Kruskal–Wallis (pairwise) test was utilized to assess the statistical significance of alpha diversity. Beta diversity was performed with both qualitative (Jaccard and unweighted UniFrac) and quantitative (Bray–Curtis and weighted UniFrac) distance metrics at sample depth of 1,000 reads. In this process, QIIME2 *diversity* plugin was employed. Statistical significance among different groups was

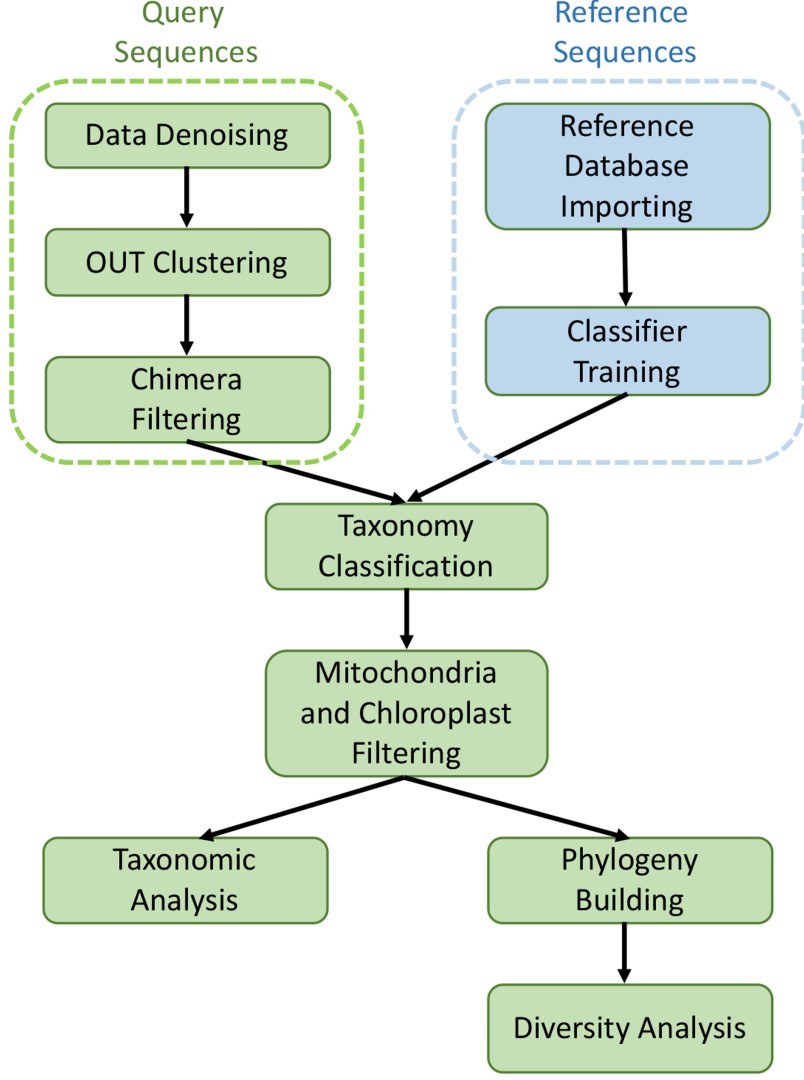

**Figure 1 Graphical workflow of metagenomic analysis in our study.**

evaluated by permutation-based ANOVA (PerMANOVA) test (*Anderson, 2005*) with 999 permutations (beta-group-significance command in *diversity* plugin). Principal coordinates analysis plots (PCoA) were generated by *Emperor* tool of QIIME2 to explore the bacterial community structure. The bar plots showing taxonomy levels were generated by QIIME2 *taxa* plugin. The metagenomic analysis workflow is shown in Fig. 1.

The Venn diagram was generated with a WWW-based tool to calculate the intersection(s) of the list of elements that in this study was represented by the list of genera of bacteria found in each climate condition and species. The graphical output is in the form of a Venn/Euler diagram.

## Seed germination test

Seeds of 19 cool-season cultivars were placed in Petri dishes containing 25 ml 1.5% agar. All Petri dishes were kept in a growth chamber at 28 °C. Seed germination was observed

every 24 h until no more seed germinated. Seed germination rates and time for each sample were calculated and correlation analysis was performed and visualized with python-based libraries *SciPy* (V0.19.1), *pandas* (V0.22.0), *seaborn* (V0.9.0) and *matplotlib* (V2.2.3).

## RESULTS

### Sequence analysis

In total, 7,405,226 sequences (about 274,368 sequences per sample) were generated by Illumina MiSeq sequencing and imported into QIIME2 pipeline suite for analysis. After being denoised and dereplicated with *dada2* pipeline, the remaining high-quality sequences were clustered into 310 OTUs that had an average length of 427 bp, ranging from 267 bp to 440 bp. After the removal of mitochondrial and chloroplast genomes, a total of 247 OTUs were used to represent the bacterial profile of turf seeds samples (Table S2).

### Diversity of bacterial endophytes associated with turf seeds from LM and HM climates

The bacterial community associated with turf seed samples was composed of five phyla, eight classes, 21 orders, 37 families and 69 genera. The bacterial community vectored by seeds produced in HM climates covered all discovered taxonomies, with 6,644 sequences/sample. However, seeds from LM climate only hosted part of them, four phyla, eight classes, nine orders, 10 families and 15 genera, with 2,821 sequences/sample.

Regardless of the climate and turf species, bacterial communities at phylum level were dominated by Proteobacteria (51%) and Bacteroidetes (40%). Proteobacteria took 89% and 50% of the bacterial community on HM and LM climates seeds, respectively. Bacteroidetes was abundant in HM climate seeds (39%) but not LM climate seeds (2%). Actinobacteria (6%) and Firmicutes (3%) also comprised a portion of the bacterial community and exhibited no significant difference between the two climate types.

Both HM and LM seeds shared some of the most abundant bacterial classes, that is, Actinobacteria, Bacteroidia, Bacilli, Alphaproteobacteria and Gammaproteobacteria (Fig. 2). Compared to LM seeds, seeds from HM environment was richer in terms of Faith's phylogenetic diversity (Fig. 3; Table S3). At class level, Actinobacteria, Bacteroidia and Bacilli had the same portion as phyla Actinobacteria, Bacteroidetes and Firmicutes, respectively (Table 1). Alphaproteobacteria and Gammaproteobacteria were the two classes within phylum Proteobacteria. Alphaproteobacteria took 6% and 18% of the bacterial community of LM and HM climate seeds, respectively. Gammaproteobacteria was 83% and 32% for LM and HM climate, respectively.

At genus level, LM seeds harbored a significantly higher percentage of *Massilia* ($p = 0.013$), *Pantoea* ($p = 0.060$) and *Pseudomonas* ($p = 0.045$) compared to HM seeds (Table 1). In contrast, HM seeds harbored more of *Flavobacterium* ($p < 0.001$), *Chryseobacterium* ($p < 0.001$), *Pedobacter* ($p < 0.001$), *Sphingomonas* ($p = 0.035$) and *Erwinia* ($p = 0.122$) (Table 1).

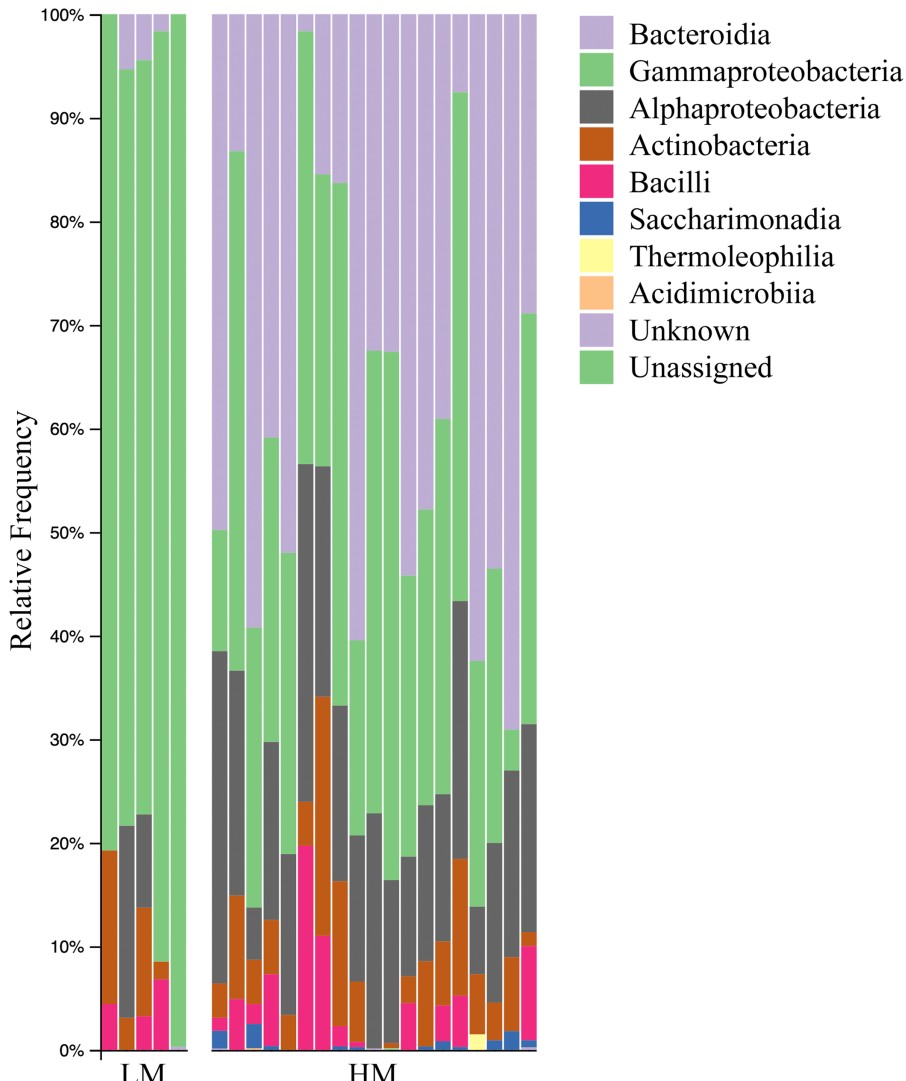

**Figure 2 Bar plot analysis illustrating the relative abundance and distribution of the OTUs assigned to class-level taxonomy.** LM, low moisture; HM, high moisture.

Principal coordinates analysis plots separated bacterial communities associated with turf seeds by climate (Fig. 4). Also, the PERMANOVA test showed a significant difference between the two groups ($p = 0.002$, Table S4). However, no significant correlation was detected between different species of turf seeds and their bacterial profile (*Festuca rubra* vs. *Lolium arundinacea*, $p = 0.101$; *F. rubra* vs. *L. perenne*, $p = 0.109$; *L. arundinacea* vs. *L. perenne*, $p = 0.204$, Table S4).

LM seeds didn't bear any unique bacteria genus that HM seed didn't (Fig. 5A; Table S5). However, HM seeds harbored 55 genera that LM seeds didn't. Seeds of three turf species shared four phyla and 27 genera, including five genera that were uncultured or unknown species from either Bacterioidetes or Patescibacteria (Fig. 5B; Table 2).
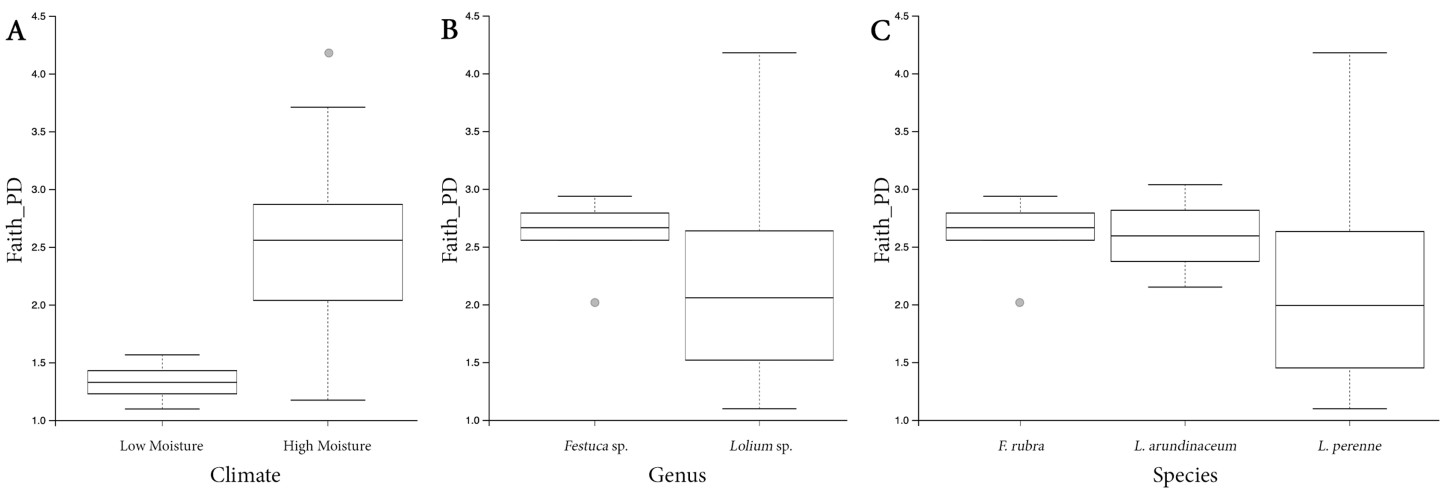

**Figure 3** Box plots depicting the Faith's phylogenetic diversity for different climate conditions (A), different genera (B) and different species (C).

## Correlation of seed germination and bacterial endophyte composition associated with turf seeds

Bacterial groups at different taxonomic ranks correlated with the seed germination rate and time (Figs. 6 and 7). Among the five phyla that we discovered through diversity analysis, Proteobacteria correlated positively with the seed germination rate ($p$ = 0.028) and negatively with the seed germination time ($p$ = 0.016). Phylum Actinobacteria also showed a negative correlation with the seed germination time ($p$ = 0.040) but not a significant correlation with germination rate ($p$ = 0.120). Another phylum, Firmicutes, showed correlation with germination ($p$ = 0.109) rate and germination time ($p$ = 0.069), but this was not statistically significant. However, the abundance of Bacteroidetes was negatively associated with the seed germination rate ($p$ = 0.008), and positively associated with the seed germination time ($p$ = 0.002).

At class level, Bacilli and Gammaproteobacteria were groups showing exactly the same correlation as phyla Firmicutes and Proteobacteria, respectively (Figs. 6 and 7). Also, the abundance of Bacteroidia and Gammaproteobacteria showed a similar correlation to phylum Bacteroidetes and Proteobacteria.

At family level, seed germination rate was positively related to the abundance of bacteria from families Microbacteriaceae ($p$ = 0.090), Paneibacillaceae ($p$ = 0.109) and Pseudomonadaceae ($p$ = 0.138), and negatively associated with the abundance of bacteria from Rhizobiaceae ($p$ = 0.014), Sphingobacteriaceae ($p$ = 0.005), and Weeksellaceae ($p$ = 0.033). As expected, seed germination time also correlated negatively with the abundance of these bacterial families. Seed germination time was positively associated with Rhizobiaceae ($p$ = 0.004), Sphingobacteriaceae ($p$ = 0.002), and Weeksellaceae ($p$ = 0.008), but negatively with Microbacteriaceae ($p$ = 0.032) and Paneibacillaceae (0.069) and Pseudomonadaceae ($p$ = 0.049).

At genus level, the abundance of *Rhizobium, Chryseobacterium* and *Pedobacter* was negatively associated with germination rate ($p$-value 0.041, 0.033 and 0.004, respectively)

**Table 1 Composition of bacterial community from LM and HM climate seeds.**

| Taxa level | Taxa name | Average percentage | | |
|---|---|---|---|---|
| | | LM (%) | HM (%) | Combined (%) |
| Phylum | Actinobacteria** | 6 | 7 | 6 |
| | Bacteroidetes[†] | 2 | 39 | 40 |
| | Firmicutes** | 3 | 4 | 3 |
| | Patescibacteria | – | <1 | <1 |
| | Proteobacteria[†] | 89 | 50 | 51 |
| Class | Actinobacteria** | 6 | 6 | 6 |
| | Bacteroidia | 2 | 39 | 40 |
| | Bacilli** | 3 | 4 | 3 |
| | Saccharimonadia | – | <1 | <1 |
| | Alphaproteobacteria[†] | 6 | 18 | 15 |
| | Gammaproteobacteria[†] | 83 | 32 | 36 |
| Order | Micrococcales** | 6 | 5 | 5 |
| | Cytophagales | <1 | 2 | 2 |
| | Flavobacteriales[†] | <1 | 11 | 9 |
| | Sphingobacteriales[†] | 1 | 26 | 21 |
| | Bacillales** | 3 | 4 | 4 |
| | Saccharimonadales | – | <1 | <1 |
| | Rhizobiales | – | 3 | 2 |
| | Sphingomonadales[†] | 6 | 15 | 13 |
| | Betaproteobacteriales[†] | 30 | 13 | 17 |
| | Enterobacteriales[†] | 24 | 12 | 15 |
| | Pseudomonadales[†] | 29 | 6 | 11 |
| Family | Microbacteriaceae** | 6 | 5 | 5 |
| | Hymenobacteraceae | <1 | 1 | 1 |
| | Flavobacteriaceae[†] | <1 | 3 | 3 |
| | Weeksellaceae[†] | <1 | 7 | 6 |
| | Sphingobacteriaceae[†] | 1 | 26 | 21 |
| | Paenibacillaceae** | 3 | 4 | 4 |
| | Rhizobiaceae[†] | – | 2 | 1 |
| | Sphingomonadaceae[†] | 6 | 15 | 13 |
| | Burkholderiaceae[†] | 30 | 13 | 17 |
| | Enterobacteriaceae[†] | 24 | 12 | 15 |
| | Pseudomonadaceae[†] | 29 | 6 | 11 |
| Genus | *Curtobacterium*** | 2 | 2 | 2 |
| | *Hymenobacter* | <1 | 1 | 1 |
| | *Flavobacterium*[†] | <1 | 3 | 3 |
| | *Chryseobacterium*[†] | <1 | 7 | 6 |
| | *Mucilaginibacter* | – | <1 | <1 |
| | *Pedobacter*[†] | 1 | 25 | 20 |
| | *Paenibacillus*** | 3 | 4 | 4 |

| Taxa level | Taxa name | Average percentage | | |
|---|---|---|---|---|
| | | LM (%) | HM (%) | Combined (%) |
| | *Rhizobium** | – | 2 | 1 |
| | *Sphingomonas*[†] | 6 | 15 | 13 |
| | *Duganella*[†] | 5 | 3 | 3 |
| | *Massilia*[†] | 25 | 9 | 12 |
| | *Erwinia*[†] | <1 | 3 | 3 |
| | *Pantoea*[†] | 23 | 9 | 12 |
| | *Pseudomonas*[†] | 29 | 6 | 11 |
| Unassigned | | <1 | <1 | <1 |

**Notes:**
[*] *Rhizobium* group also includes *Allorhizobium*, *Neorhizobium*, and *Pararhizobium*.
[**] Bacterial groups without significant difference between LM and HM.
[†] Bacterial groups with significant difference between LM and HM.

but positively with germination time ($p$-value 0.015, 0.008 and 0.001, respectively). But the abundance of *Pseudomonas* was positively related with germination rate ($p = 0.138$) but negatively associated with the germination time ($p = 0.049$), although the correlation was not significant.

## DISCUSSION

### A complex bacterial community associated with turf seeds

Compared to LM seeds, HM seeds harbored a more diverse bacterial community with many more bacterial cells (i.e., a higher bacterial load), as there were more sequences associated with HM seeds. The more individuals hypothesis, which predicts that communities with more individuals will have more species (*Storch, Bohdalková & Okie, 2018*), can explain the higher diversity associated with HM seeds. This result is similar to previous studies on soils indicating that moisture controls the structure and function of the soil microbial community (*Brockett, Prescott & Grayston, 2012*; *Griffiths et al., 2003*; *Steven et al., 2013*). Water availability in soil controls bacterial composition (*Zeglin et al., 2011*). Similarly, relatively HM will favor growth and replication of bacteria on seeds, while LM conditions will suppress development of the bacterial community associated with seeds.

Both LM and HM seeds vectored a large number of bacteria and shared some groups. For example, the abundance of *Curtobacterium* spp. was similar in both LM and HM seeds (LM 2%, HM 2%). *Curtobacterium* is a Gram-positive endophytic bacterial genus in rice seeds (*Oryza sativa*), field-grown tall fescue (*L. arundinacea*) and *Noccaea goesingensis* (*De Los Santos et al., 2015*; *Mano et al., 2006*; *Ruiz et al., 2011*). Some *Curtobacterium* strains provide host growth promotion and pathogen antagonistic effects (*De Los Santos et al., 2015*; *Ruiz et al., 2011*). *Paenibacillus* was the only genus from phylum Firmicutes in both LM and HM seeds (LM 3%, HM 4%). *Paenibacillus* have been isolated from many plants and shown to produce IAA, solubilize phosphate and inhibit the growth of phytopathogens (*Aswathy et al., 2013*; *Diaz Herrera et al., 2016*; *Ruiz et al., 2011*; *Rybakova et al., 2015*).
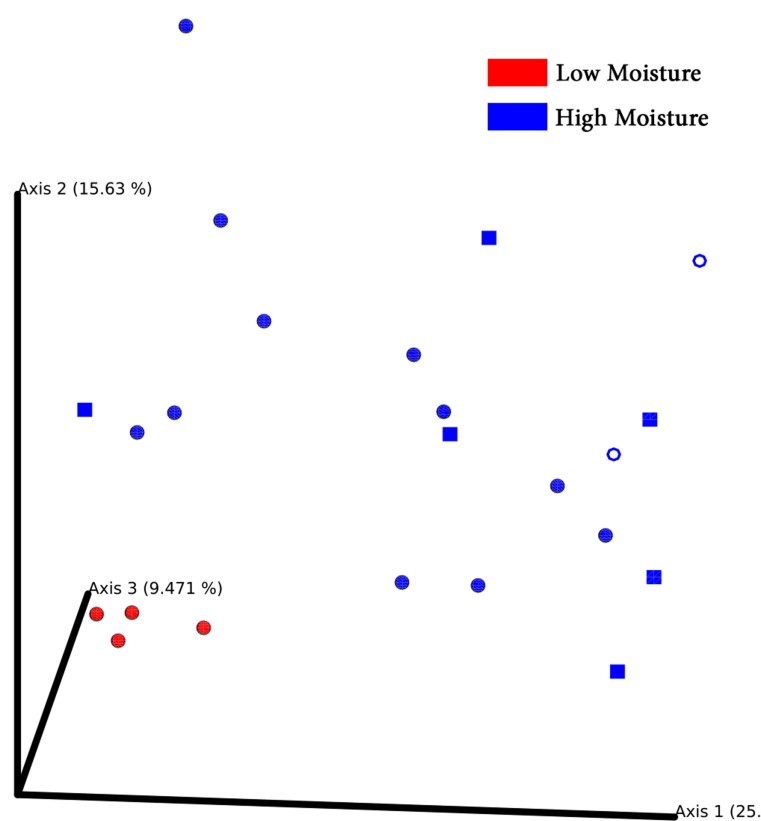

**Figure 4 PCoA Emperor plots based on Bray–Cuitis diversity matrix.** Samples are scattered concerning their bacterial community. Climates are represented by different colors: red-low moisture; blue-high moisture. Species were represented by different shapes: ring-*Loium arudinacea*; sphere-*Lolium perenne*; square-*Festuca rubra*.

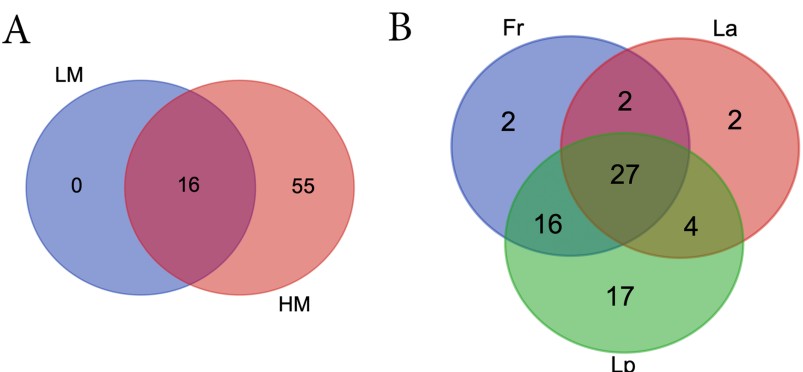

**Figure 5 Venn diagrams showing the number of shared bacterial genera between different climate (A) and among different turf species (B).** LM, low moisture; HM, high moisture; Fr, *Festuca rubra*; Lp, *Lolium perenne*; La, *Loium arudinacea*.

Two of the genera, *Mucilaginibacter* and *Rhizobium*, were found in HM seeds but not LM seeds. *Mucilaginibacter* spp. can promote plant growth and produce extracellular polysaccharides (*An et al., 2009*; *Lee et al., 2013*; *Madhaiyan et al., 2010*; *Mannisto et al., 2010*).

**Table 2 Seed-vectored bacterial genera shared *Loium arudinacea*, *Lolium perenne*, and *Festuca rubra*.**

| Phylum | Genus |
|---|---|
| Actinobacteria | *Curtobacterium* |
| | *Sanguibacter* |
| Bacteroidetes | *Chryseobacterium* |
| | *Dyadobacter* |
| | *Flavobacterium* |
| | *Mucilaginibacter* |
| | *Pedobacter* |
| | *Sphingobacterium* |
| | *Spirosoma* |
| | Uncultured Sphingobacteriaceae |
| | Unknown Sphingobacteriaceae |
| Patescibacteria | Uncultured bacterium |
| | Uncultured *Sphingobium* sp. |
| | Unknown Saccharimonadales |
| *Proteobacteria* | *Rhizobium*[*] |
| | *Aureimonas* |
| | *Brevundimonas* |
| | *Devosia* |
| | *Duganella* |
| | *Erwinia* |
| | *Massilia* |
| | *Novosphingobium* |
| | *Pantoea* |
| | *Pigmentiphaga* |
| | *Pseudomonas* |
| | *Sphingomonas* |
| | *Verticia* |

**Note:**
[*] *Rhizobium* group also includes *Allorhizobium*, *Neorhizobium*, and *Pararhizobium*.

*Rhizobium* together with *Allorhizobium*, *Neorhizobium*, and *Pararhizobium*, composed 2% of the bacterial community on HM seeds. These genera comprise well-studied bacteria that promote growth of plants and nodulate legumes to fix nitrogen (*Datta & Basu, 2000*; *Gutierrez-Zamora & Martınez-Romero, 2001*; *Kiers et al., 2003*; *Yanni et al., 1997*).

At the phylum level, LM seeds hosted more Gammaproteobacteria than HM seeds. Several genera within Gammaproteobacteria contributed to these results, that is, *Duganella*, *Massilia*, *Pantoea*, and *Pseudomonas*. However, these bacteria were still found on a large portion of the HM seeds. *Duganella* spp. can suppress the growth of plant pathogens (*Cretoiu et al., 2013*; *Haack et al., 2016*). *Massilia* is a root-colonizing bacterial genus with the ability to degrade chitin (*Adrangi et al., 2010*; *Faramarzi et al., 2009*; *Ofek, Hadar & Minz, 2012*). *Pantoea* spp. promote plant growth and tolerance of

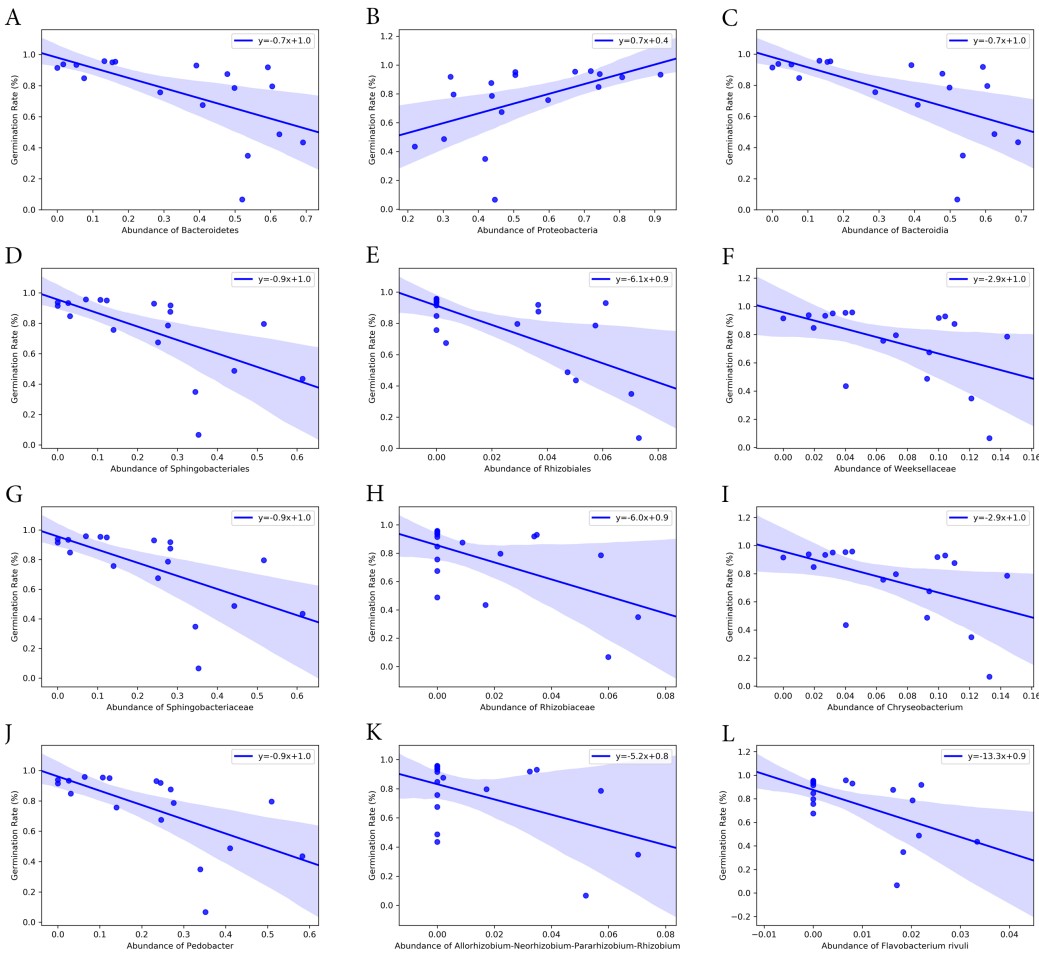

**Figure 6 Correlation of seed germination rate with abundance of bacteria groups at different taxonomy levels.** (A) and (B) Phylum level. (C) Class level. (D) and (E) Order level. (F)–(H) Family level. (I)–(K) Genus level. (L) Species level.

environmental stresses (*Chen et al., 2017*; *Feng, Shen & Song, 2006*; *Ferreira et al., 2008*; *Gond et al., 2015*). *Pseudomonas* contains many endophytic bacterial strains that benefit hosts by producing IAA, producing biocontrol lipopeptides, and solubilizing phosphate (*Oteino et al., 2015*; *Prieto & Mercado-Blanco, 2008*; *Suzuki, He & Oyaizu, 2003*).

Some bacterial genera were more abundant in HM seeds than LM seeds, including *Flavobacterium*, *Chryseobacterium*, *Pedobacter*, *Sphingomonas* and *Erwinia*. Most of the bacteria comprised a very small portion of the bacterial community of LM seeds, but *Sphingomonas* made up 6%. *Flavobacterium* sp. has been found to promote plant growth and provide biocontrol activity to the hosts (*Kolton et al., 2016*; *Soltani et al., 2010*). *Chryseobacterium* spp. were also shown to be plant growth promoting bacteria (*Dardanelli et al., 2009*; *Gutiérrez Mañero et al., 2003*). Although *Pedobacter* has not been found to promote growth of plants, it can induce the production of antimicrobial compounds by *Pseudomonas fluorescens* Pf0-1 (*Garbeva et al., 2011*). *Sphingomonas* is an alphaproteobacterial genus containing strains that produce IAA and provide nutrients to hosts (*Okunishi et al., 2005*; *Ruiz et al., 2011*). *Erwinia* spp. have also been identified as

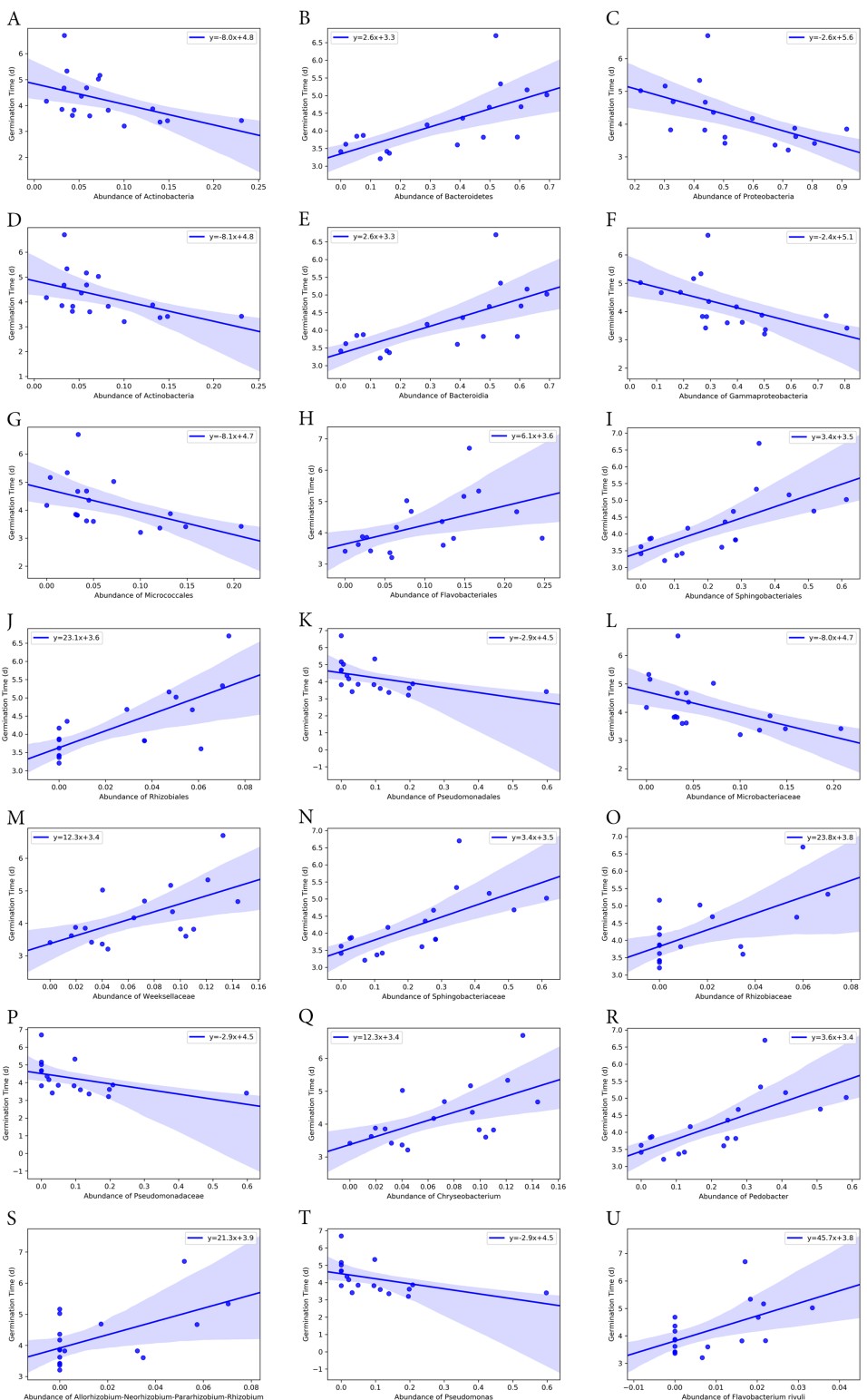

**Figure 7 Correlation of average seed germination time with abundance of bacteria groups at different taxonomy levels.** (A)–(C) Phylum level. (D)–(F) Class level. (G)–(K) Order level. (L)–(P) Family level. (Q)–(T) Genus level. (U) Species level.

endophytes in some plant species (*Verma, 2019*). However, genus *Erwinia* is well-known to contain many plant pathogenic species.

In total, the above genera together comprised 89% and 95% in HM and LM seeds, respectively. Some of the bacterial genera include plant pathogens, for example, *Ewrinia* and *Pseudomonas*. However, most of the bacteria are known to contain mainly plant growth promoting rhizobacteria, for example, *Paenibacillus, Pseudomonas, Rhizobium, Pantoea*. The seed microbes are important because they stimulate seedling development, increase stress tolerance in seedlings and protect seedlings from disease (*Verma et al., 2017*, *2018*; *White et al., 2018*, *2019*). Thus, a more diverse bacterial community on HM seeds may provide hosts with more microbial resources to utilize for plant development and stress tolerance.

In the study, the samples of LM and HM seeds had unequal sizes (LM: 4; HM: 19), which could create a bias in our final result. However, HM seeds vectored a more diverse bacterial community with significantly more bacteria cells (Fig. 3; Table S3). Also, the different abundances between bacterial groups were statistically significant.

### The bacterial community affected seed germination and growth

Seeds from HM climates tended to show slower germination and reduced seedling growth rates. These seeds vectored a denser and more diverse community of bacteria, which may benefit seedlings but not without a cost. We hypothesize that the higher microbial load competes with seedlings, which slows germination and development of the host. This nutritional cost may result in slower seed germination and seedling development rates. Seed growers have observed that seed from HM climates seems to establish better with reduced damping-off disease compared to seed from LM climates (W. Meyer, 2016, unpublished data). While, seeds with richer and denser microbiomes grow slower initially, they may be better protected from soil borne pathogens than seeds with less developed microbiomes.

Seeds that have formed in LM situations, or where the natural microbiome has otherwise been damaged, could be remediated through application of microbes in seed coatings (*Pedrini et al., 2017*). Coating formulations with the correct microbes at the optimal concentrations could result in better fitness of seeds and seedlings.

## CONCLUSIONS

We surveyed the bacterial community associated with seeds of several species of cool-season turfgrasses and identified the dominant bacterial groups of the communities at different taxonomic levels. Regardless of the moisture level during seed production and species of seeds, the core bacterial community included many PGPB strains. Seeds produced in HM conditions maintained a denser and more diverse bacterial community than seeds produced in LM conditions. This seed microbiome may help seedlings tolerate stress but may also compete with seedlings for nutrients and slow early seedling growth.

## ACKNOWLEDGEMENTS

The authors appreciate the support of Steve Reid (DLF Pickseed USA) for providing the seed resources. The authors also thank the Department of Plant Biology, Rutgers University, NJ for research facilities.

### Funding

This work was supported by USDA-NIFA Multistate Project W4147, the Rutgers Turf Science Center, and the New Jersey Agricultural Experiment Station. The funders had no role in study design, data collection and analysis, decision to publish, or preparation of the manuscript.

### Grant Disclosures

The following grant information was disclosed by the authors:
USDA-NIFA Multistate Project: W4147.
Rutgers Turf Science Center.
New Jersey Agricultural Experiment Station.

### Competing Interests

The authors declare that they have no competing interests.

### Author Contributions

- Qiang Chen conceived and designed the experiments, performed the experiments, analyzed the data, prepared figures and/or tables, authored or reviewed drafts of the paper, and approved the final draft.
- William A. Meyer conceived and designed the experiments, authored or reviewed drafts of the paper, and approved the final draft.
- Qiuwei Zhang analyzed the data, prepared figures and/or tables, and approved the final draft.
- James F. White conceived and designed the experiments, authored or reviewed drafts of the paper, and approved the final draft.

### Data Availability

 The raw data is available on at FigShare: Chen, Qiang; Meyer, William A.; Zhang, Qiuwei; White, James F. (2019): turf_seeds_raw_data. figshare.
DOI 10.6084/m9.figshare.9745061.v1.

### Supplemental Information

Supplemental information for this article can be found online at http://dx.doi.org/10.7717/peerj.8417#supplemental-information.

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
