# Peer review of "S rRNA metagenomic analysis of the bacterial community associated with turf grass seeds from low moisture and high moisture climates"

_PeerJ, doi:10.7717/peerj.8417_

## Round 0.1 · original submission · Major Revisions

The role humidity plays in structuring the bacteria community associated with seeds interests me and I believe will interest readers too, particularly as it relates to best practices for seed production and storage. As I understand it, bacterial communities were compared between seeds produced at different locations. I think an experiment where the humidity is manipulated would be more powerful. I suggest you justify your design in Introduction and discuss its limitations in Discussion. The unequal sample size, only a few seeds from low moisture climates, could create a bias that should be also be discussed.

As submitted, Discussion mostly repeats Results and notes where others have seen similar trends. I suggest adding a discussion of how the higher diversity associated with moisture relates to species-energy theory. That is more individuals correlates to higher diversity.

The correlation between abundance of particular bacteria and seed germination interests me too, but those trends, shown in Figure 6, are not presented in Abstract. I suggest revising Abstract to include that experiment and including, in Discussion, how this points to other applications, such as seed coatings.

I list numerous, but not comprehensive, style and format edits below that should be considered before we recruit reviewers.

Line 17. I do not see the need for the word “however” here. It seems self-evident that moisture influences bacterial community structure. I suggest adding a justification for the work (“little is known..”).
Line 19. Overall, the text is wordy. For example, revise to “…formation of this microbial community. In this study…”
Line 20. You need to mention that you looked at seeds from three grass species here and present the methods used in seed germination experiment too.
Line 22-23. Again, wordy. I suggest replacing “A significant..seeds.” with “Climate structured the bacterial community associated with seeds.”
Line 23-26. Use parallel sentence construction to compare treatments. I suggest replacing “On/in LM..dominated by Bacteroides” with “LM seeds vectored mainly Proteobacteria. HM seeds vectored a more diverse bacterial community that included Proteobacteria (50 %) and Bacteroides (39%).
Line 28. Given the precision of amplicon libraries the number of significant figures should be zero (13% not 12.71%) here and throughout.
Line 32. I am not sure what the word “nevertheless” adds here.
Line 35-36. Separate phrases, like “in terms of …climates”, with commas.
Line 38-42. Again, wordy. I suggest revising to “..may benefit seedlings by helping them tolerate stress but this dense microbial community may compete with seedlings for nutrients.”
Line 47. Here and throughout, avoid empty phrases like “A study on” and “have been shown to (line 49).” These fillers only create confusion, as they focus the reader on the scientist and not the subject of the science. I suggest that you state that the root-associated microbes influence plant phenotypes and then discuss how these microbes can come from soil or seeds. We know a little about what structures the soil microbiome (pH…) but little about the seed microbiome.
Line 51. As above. Delete “As shown in the study by” and revise to “Cruz et al. (year) showed..
Line 52. The phrases “Some studies revealed that” suggests some studies revealed something else. As above, delete these confusing fillers.
Line 62-64. The sentence “It thus…disease.” adds little. Delete it.
Line 65. Delete phrase “have been shown to” and “have been found to (line 71)”
Line 80. The last sentence of Introduction should state the key result observed in the study.
Line 88. Add a space between numbers and units (750 mm) here and throughout.
Lin 106 – 109. Combine this into one sentence. I suggest “..library was paired-end sequenced (2 x 300 bp) on an..”
Line 162. I think there can only be one dominant. Bacteroides was also abundant.
Line 165. Do not send the reader away to look at a Figure, in the case Figure 2, without first telling them what the figure shows.
Line 181-185. This paragraph belongs in Methods.
Line 200. The first sentence in each paragraph in Results should be a topic sentence that states the key observation. I suggest a statement that the seed microbiome corresponded to seed germination.
Line 232. Delete “This suggests…grasses.”
Line 235. Delete “Also” and revise to “water availability controls bacterial composition.”
Line 236. Delete “based on 16S rRNA study”
Line 236-237. Wordy. Revise to “Bacteria depend on water..”
Line 241. Not sure why “However” is here.
Line 243. Delete “that has been demonstrated to,” “have been found to (line 246) and “has been isolated in previous studies and demonstrated to.”
Line 248. Delete “In previous studies.”
Line 264. The phrase “was suggested to” suggests those results were not supported by subsequent work. Revise to “Dugenella sp can suppress…”
Line 267. Delete “was demonstrated to”
Line 296. Let the reader decide what interests them.
Line 302. Wordy. Revise to “Seed growers have observed that seed..”
Line 303. Revise to “reduced”
Line 317. See comment about lines 38-42.
Line 320. Delete “Additional studies …here.”
Line 393. Please carefully proofread references. Cap journal titles (Biological Conservation, Advance in Microbial Ecology (line 511)) and book titles to (Advances in Endophytic Research (line 588)).
Line 406. Either use abbreviations for journal titles or not (FEMS Microbiology Letters).
Line 541. I think “in vitro” should be italicized but Shaik and Thomas didn’t either, so who knows. The word “activation” should be all lower case.
Line 573. Italicize genus and species.

---

## Round 0.2 · Minor Revisions

The reviewers suggested minor changes, which I believe can be addressed. I also suggest that you consider revising to make the text more concise. For example, in Abstract this sentence “Previous work on grass seeds has shown that seeds carry some microbes on their surfaces and these microbes have been shown to become endophytic in seedlings and promote seedling growth after germination (line 20-22)” contains phrases that add no content. I suggest revising to “Grass seeds carry microbes on their surfaces that become endophytic in seedlings and promote seedling growth.”

Michael LaMontagne
UH - Clear Lake

·

Basic reporting

The work is very interesting from the point of view of microbial diversity in seeds, however the DNA of microorganisms that live on the surface and as endophytes of the seeds was generally obtained.
Which is very important because of the variable they are evaluating, because the abiotic factors modify the microbial diversity.

Experimental design

no comment'

Validity of the findings

In the discussion, enrich yourself with the effect of biotic factors on microbial diversity; as well as how it affects the development of diseases in low moisture and high moisture climates

Additional comments

I recommend that the authors mention in the discussion how the low
moisture and high moisture climates, could affect the microorganisms on the surface of the seeds; in the same way to endophyte microorganisms

Reviewer 2 ·

Basic reporting

Should improve figure 2 quality.

Experimental design

During the MS, the community studied in this work is the one associated to seeds. But the author uses the “bacterial communities on/in turf seeds”. I suggest rephrasing this to “bacterial communities associated to turf seeds” along the MS

In line 95, there is no information regarding the seed’s treatment. The seeds were washed with water, alcohol or any other agent?
When reading results or supplemental material, it is said the sample n (LM: 4; HM: 19). As this is relevant data, should be clearly written in methods.
Primers listed in line 106 were previously used or designed in this work? Should cite previous work or make the reader know if it has not been previously used.
In line 150 the versions of the programs are required.
The information within the paragraph 181-186 lines, is in any table? If not, should be for smooth reading.
The more sequenced named in line 234, are statistical supported? In this case it may be said in the sentence.

Validity of the findings

The paragraph between lines 293-296 assures that HM sample is more diverse than LM sample. I don’t think that statement is correct given the unequal n samples. The n samples should be normalized to the same, should be 4 for both samples for statistical studies. Do not seem fair to compare 2 so different n samples in statistics.

Annotated reviews are not available for download in order to protect the identity of reviewers who chose to remain anonymous.

---

## Round 0.3 · Minor Revisions

One reviewer suggested minor changes, which I believe can be addressed; however, the manuscript still needs extensive revision for style. In my opinion, my suggestion to write more concisely was not followed. I provide specific edits below.

Line 21-23. Avoid passive voice. Revise to “We hypothesized that incomplete development of the microbiome associated with the surface of seeds produced in dry climates reduces the performance of seeds.”
Line 24. Delete “In this study”
Line 28. Move seed germination methods here. “…V3-V4 regions. We performed seed germination tests and analyzed the correlations between .. different taxonomy ranks. Climate…”
Line 31. The phrase “.. took similar percentages associated with both LM and HM..” is awkward and adds little. Delete “Also..seeds.”
Line 34-35. Use parallel structures to present parallel ideas. Revise to “…conditions. Massilia, Pantoea and Pseudomonas dominated LM seeds. Pedobacter dominated HM seeds.”
Line 50-52. Write more concisely. Revise to “Plants bear numerous microbes that influence their nutrition, development, stress responses and phenotypes (White et al., 2014; White et al., 2018
Henning et al..”
Line 54-70. This section is wordy. Revise to - “Plant-associated microbes generally come from the surrounding environment; however, Johnston-Monje and Rhizada (2011) showed that corn
seeds vectored a diverse array of microbes. Further, Johnston-Monje et al. (2016) found that the rhizospheres of corn seedlings were composed of microbes that originated both from seeds and
bacteria recruited from soils. These seed-vectored bacteria can influence germination and share a mutualistic association with the host seedlings (Cruz et al., 2014; Shaid & Thomas, 2019; Somova et al., 2001; Zhu et al., 2017, Cruz et al. (2014). Without seed-vectored bacteria, seedlings can lose gravitropic response in roots, fail to develop root hairs and are more susceptible to soil-bourne pathogens (Verma et al. 2017, 2018).
Line 71. Revise to “Some turf grasses seeds internally vector fungal endophytes …”
Line 74-75. Avoid passive voice. Revise to “Turf breeders have long been employed these environmentally safe endophytes to enhance turfgrass performance and stress tolerance..”
Line 86. Mention seedling results here too.
Line 95. Do not start sentences with a number.
Line 124. Provide reference for dada2.
Line 188. Start paragraphs in Results with statements of the key result, not instructions for the reader to look at a plot. Revise to “PCoA separated bacterial communities associated with turf seeds by climate (Fig. 4).”
Line 194. As above, do not send reader away to look at Figures, until you tell them what the figure shows. Delete “The Venn.. by different species.”
Line 197. As above, delete “Figure 5B.. were shared by the” and revise to – “Seeds of three turf species shared 4 phyla and 27 genera, including five genera..”
Line 203. Replace “showed correlation” with “correlated” and “exhibited positive correlation (line 205)” with “correlated positively with.. and negatively with…”
Lne 216. Keep sentence structures parallel. Replace “As the analyses were narrowed down to family, seed…” with “At family level, seed..”
Line 218. Replace “but” with “and”
Line 224. Delete “Finally” and start a new paragraph with “At genus level,..”
Line 236. Avoid passive voice. Revise to “The More Individuals Hypothesis, which predicts that communities with more individuals will have more species (Storch, Bohdalková & Okie, 2018), can explain the higher diversity associated with HM seeds” Also, how is this not Species Area/Energy theory?
Line 239. Replace “moisture is a major factor influencing the structure” with “moisture controls the structure.. Steven et al., 2013, Zeglin et al., 2011). Similarly..”
Line 255. Avoid phrases like “was demonstrated to,” “has been found to (line 276)”, and “was also shown to be (line 277), which add no content. Replace “Mucilaginibacgter sp. was demonstrated to be plant-growth-promoting bacteria produce” with “Mucilaginibacgter sp. can promote plant growth and produce…”
Line 267. Revise to “Pantoea sp. promote plant growth and tolerance of environmental stress…”
Line 280. Replace “…plants, it has been demonstrated to induce..” with “plants, it can induce…”
Line 301-306. This section is wordy. Revise to “…growth rates. These seeds vectored a more diverse and dense community of bacteria, which may benefit seedlings but not without a cost. We hypothesize that the higher microbial load competes with seedlings, which slows germination and development of the host.”
Line 311. Also wordy. Revise to “Seeds that have formed in low moisture situations, or where the natural microbiome has otherwise been damaged, could be remediated through application of microbes in seed coatings (Pedrini et al., 2017).”
Line 321 -328. Wordy. Revise to “Regardless of the moisture level during seed production and species of seeds, the core bacterial community included many PGPB strains. Seeds produced in high moisture conditions maintained a more diverse and denser bacterial community than seeds produced in low moisture conditions. This seed microbiome may help them tolerate stress but may also compete with seedlings for nutrients and slow early seedling growth.”

·

Basic reporting

IIn general, the manuscript is well structured, so the methodology proposed is consistent with the objective of the research. As well as the research topic is of great interest, about the diversity of microorganisms associated with seeds and their role in tolerance to biotic and abiotic factors, so I recommend that it be accepted for publication

Experimental design

The authors mention in the discussion that the greater diversity of microorganisms can help to tolerate stress, however they do not refer to any specific genre, which is why I suggest enriching the discussion by relating the presence of bacterial genera with stress toleranc

Validity of the findings

No cmment

Additional comments

How can we differentiate which microorganisms are found on the surface of the seed from those that are endophytes?

Reviewer 2 ·

Basic reporting

no comment

Experimental design

no comment

Validity of the findings

no comment

Additional comments

The authors have incorporated the suggestions that the reviewers have indicated.
The ease of reading the MS has improved, as the suggested changes had been added

---

## Round 0.4 · accepted · Accept

I appreciate your careful consideration of the reviewer's and my comments.

Michael